# Public Health Response to the SARS-CoV-2 Pandemic: Concern about Ultra-Processed Food Consumption

**DOI:** 10.3390/foods11070950

**Published:** 2022-03-25

**Authors:** Sara De Nucci, Roberta Zupo, Fabio Castellana, Annamaria Sila, Vincenzo Triggiani, Giuseppe Lisco, Giovanni De Pergola, Rodolfo Sardone

**Affiliations:** 1Unit of Data Sciences and Technology Innovation for Population Health, National Institute of Gastroenterology IRCCS “Saverio de Bellis”, Research Hospital, Castellana Grotte, 70013 Bari, Italy; sara.denucci@irccsdebellis.it (S.D.N.); fabio.castellana@irccsdebellis.it (F.C.); annamaria.sila@irccsdebellis.it (A.S.); rodolfo.sardone@irccsdebellis.it (R.S.); 2Unit of Geriatrics and Internal Medicine, National Institute of Gastroenterology IRCCS “Saverio de Bellis”, Research Hospital, Castellana Grotte, 70013 Bari, Italy; gdepergola@libero.it; 3Section of Internal Medicine, Geriatrics, Endocrinology, and Rare Disease, Interdisciplinary Department of Medicine, School of Medicine, University of Bari, 70124 Bari, Italy; vincenzo.triggiani@uniba.it; 4Unit of Endocrinology, Metabolic Disease & Clinical Nutrition, Hospital “A. Perrino”, 72100 Brindisi, Italy; g.lisco84@gmail.com

**Keywords:** ultra-processed foods, dietary habits, eating behavior, SARS-CoV-2, pandemic, review

## Abstract

**Introduction:** There is scientific consistency in the concept of ultra-processed foods (UPFs) as a descriptor of an unhealthy diet. The most recent literature points to troubling evidence that policies adopted to address the SARS-CoV-2 pandemic may have contributed to diverting eating habits toward a poorer diet. Considering the historically unique SARS-CoV-2 pandemic lockdown scenario, and the health burden imposed by UPFs on human health, it is critical to investigate how the epidemic has influenced UPF intake directly. Reviewing the literature, we aimed to assess the changes in the consumption of UPFs during the pandemic lockdown compared to previous habits in the general population. **Methods:** Consulting six databases, we examined articles investigating the consumption of UPFs according to the NOVA classification both before the SARS-CoV-2 pandemic and during lockdowns. In total, 28 reports were included in the final analysis. **Results:** A clear trend of an increasing consumption of sweets (chocolate, candy, cookies, pastries, cakes, desserts, and confectionery, 31.75% increase vs. 21.06% decrease), packaged fatty or salty snacks (23.71% increase vs. 20.73% decrease), and baked goods (bread products, pizza, and sandwiches, 28.03% increase vs. 13.5% decrease) emerged, versus a decrease in ready-to-eat dishes (16.2% increase vs. 22.62% decrease) and ready-made meals (10.6% increase vs. 31.43% decrease), such as instant soups, canned foods, fast food, and chips, as well as sugary drinks in general (14.55% increase vs. 22.65% decrease). No trend was observed for processed meat consumption. **Conclusion:** The current pandemic scenario raises concerns about the increased consumption of UPFs, especially sweets, snacks, and baked goods, and points to an urgent need to implement policy strategies to manage the trade in these foods from a preventive perspective.

## 1. Introduction

Before the SARS-CoV-2 pandemic, 3 billion people were estimated to be unable to afford healthy food [1]. Recently, the Global Food Policy Report 2021 predicted a spike in estimates between 2020 and 2022 [2]. Indeed, among the severe health and economic consequences of the SARS-CoV-2 pandemic, and their impact on food systems and supply chains, we are experiencing an increase in low-quality, nutrient-deprived diets, inevitably leading to an increase in nutrition-related, chronic non-communicable diseases, all of which were already at all-time highs even before the pandemic. From the public health standpoint, SARS-CoV-2 has highlighted the challenges posed by our current dysfunctional food systems and policies. However, the root cause goes back to the nearly 30-year phenomenon of the “nutrition transition” [3], since the start of the widespread availability of sugar, salt, saturated fat, refined grains, and ultra-processed foods (UPFs) in the marketplace. The term “ultra-processed foods” derives from the NOVA classification, widely used in the literature, which divided foods into four categories according to the level of processing to which they are subjected. UPFs are made by performing a series of industrial steps of fractionation and reassembly of ingredients and additives, which are then wrapped in attractive packaging. Additives include those that mimic or enhance the organoleptic characteristics of food. The processes and ingredients used in the production of UPFs make them ready-to-eat, hyper-palatable, and cost-effective. The economy and attractiveness of ultra-processed foods and their massive advertising campaigns explain why they contribute to about half of the total dietary energy consumed in high-income countries, and are increasing in popularity in low- and middle-income countries. Ultra-processed foods, especially fast foods, are generally consumed in greater quantities out of home and in social situations [4]. Fast food restaurants are popular meeting places, especially for teenagers. Indeed, drinking sugar-sweetened beverages is often associated with the consumption of other foods in cafés and restaurants [5]. On the contrary, snack consumption is much more common at home or at work, especially when watching TV or playing video games. Salty snacks, sweets, candies, and sugary drinks are popular snack choices [6]. This transition has boosted the expanding epidemic of obesity and chronic lifestyle diseases, whose prevalence has increased tenfold in the last four decades, partly due to the nutritional imbalance provided by these foods at excessive consumption rates.

Of note, no age group is exempt from the effects of this transition, which is bringing about malnourishment among infants and young children, and a nutritional imbalance in aging adults, thus exacerbating the course of their physiological physical decline [7]. In practice, during the SARS-CoV-2 pandemic, governments worldwide have focused on implementing various measures to manage the viral spread. Thus, in addition to combinations of complete or partial lockdowns, travel restrictions, and immigration restrictions, the closure of restaurants and other food retailers, and the suspension or curtailment of activities and informal food businesses for weeks or months at a time, have been enforced. Along with travel and transportation restrictions, jobs were lost among food workers, farmworkers, and other sectors along food value chains, and crops were wasted, and the supply of nutritious perishable foods suffered. At the same time, massive job and income losses lowered consumer demand for relatively expensive healthy foods and fresh fruits and vegetables, which disproportionately affected low-income families. Additional factors contributing to the decline of demand for these fresh, nutrient-rich foods were temporary price increases, as well as a lack of home refrigeration and adequate storage facilities, which prevented the purchase of large quantities of perishables during lockdown periods, poor distribution of fresh foods, and also, the misperception that fresh foods (especially those of animal origin) were a risk factor for viral transmission [8].

Research efforts are currently directed toward striving to shed light on how lifestyle, the transition to smart working, and the processes mentioned above have led to changes in eating habits and patterns. Here, we previously reviewed preliminary data, and found an upward trend in the consumption of home-cooked meals, carbohydrate-rich foods, and snacks [9]. These latter are part of the so-called comfort foods—in other words, foods whose consumption provides consolation or a feeling of wellbeing due to their palatability and sweet taste releasing serotonin. For example, when considering beverages, there was increased consumption of coffee and tea during the pandemic due to their mood-modulating capacity [10]. 

Nevertheless, comfort foods also include sweets, fried foods, and snack foods, most often made by processing, and featuring tasty flavors and attractive packaging. Because of their soothing effect, depressed people tend to eat more UPFs; although, conversely, consumption of UPFs has been positively associated with depressive symptoms [11]. Top guidelines for a healthy diet strongly advise against the consumption of UPFs because of their nutritional composition: energy-dense, with a high content of saturated and trans fats, sugars, sodium, and low fiber content [12]. It is easy to understand how regular consumption of these foods may negatively impact health.

Given the historically unprecedented SARS-CoV-2 pandemic lockdown setting, and the health burden levied by UPFs on human health, it seems important to analyze how the pandemic has directly impacted UPFs intake. The present study aimed to assess the changes in the consumption of UPFs during the pandemic lockdown compared to previous habits in the general population, by means of a semi-quantitative review of recent literature.

## 2. Materials and Methods

The present is a narrative review article. Given primordial data and the topic’s novelty, we opted for a quantitative approach. Following the Preferred Reporting Items for Systematic reviews and Meta-Analyses (PRISMA) guidelines [13], we conducted individual searches in the US National Library of Medicine (PubMed), Medical Literature Analysis and Retrieval System Online (MEDLINE), EMBASE, Scopus, Ovid, and Google Scholar to find original articles investigating any differences in dietary habits during SARS-CoV-2 lockdown. In particular, our focus was to select studies examining differences in the consumption of UPFs. Our search strategy approach also accounted for the grey literature using the largest archive of preprints (https://arxiv.org/) (accessed on 20 February 2022) in the study selection process, and the http://www.opengrey.eu/ (accessed on 20 February 2022) database to access abstracts of notable conferences and other non-peer-reviewed material. To carry out the selection process, and further refine the search strategy on the UPFs, we selected studies that investigated frequencies of the consumption of all foods during the pandemic, regardless of whether NOVA was used as an assessment tool in the studies themselves. Only in a second step, we used the NOVA classification, referring to UPFs belonging to the fourth category. Briefly, the NOVA classification, first developed by Monteiro, builds on the level of food processing, and consists of four groups, i.e., unprocessed or minimally processed foods, processed culinary ingredients, processed foods, and UPFs [14]. The latter category of UPFs covers many ingredients such as food additives, colors, flavorings, and emulsifiers usually used to improve the palatability of the finished product or to mask its defects, including canned or bottled foods. As food items, UPFs are low-cost, high-fat, high-sugar, and high-salt foods, mass-produced by global multinationals, designed for an extended shelf life, and widely promoted commercially [15]. They include carbonated soft drinks, sweets, fatty or salty packaged snacks, chocolate or candy, biscuits, pastries, cakes, and sugary breakfast cereals, fruit juices, energy drinks, processed meat (sausages, hamburgers, hot dogs, and other reconstituted meat), fast food, pasta and pizza dishes, instant soups, ready-to-eat meals, desserts, and vegetable substitutes [16]. The research strategy applied to the e-sources is shown in Table 1.

As the topic is young, and the SARS-CoV-2 pandemic research period is limited, no skimming of the study population, design, or setting was applied. For the same reason, no age range was applied to the research population either. Using an Excel spreadsheet, two investigators (SDN, RZ) searched for publications separately and in duplicate, reviewing titles and abstracts of selected articles, screening full texts, and selecting articles for inclusion in this analysis.

For the present synthesis, original articles that explored dietary habits, specifically the consumption of UPFs before and during SARS-CoV-2 through online or telephone questionnaires reporting weekly or monthly food intake, met the inclusion criteria. Letters, systematic reviews, and meta-analyses were not considered. Then, all studies that provided only a snapshot of UPFs consumption during confinement were omitted in favor of those that reported before-and-after comparisons. Data were tabulated by the researchers for general information such as study design, setting, sample size and demographics (age and sex), country, dietary assessment tool, UPF diet exposure, main results, and summary of findings in terms of increased (↑) or decreased (↓) UPF consumption. Data were double-checked by a third senior researcher (FC) to address discrepancies and solve disagreements.

To foster conceptual simplification in drawing conclusions on the findings, the UPFs were grouped into eight categories [17], i.e., (1) sweets (chocolate, candies, cookies, pastries, cakes, desserts, and confectionery), (2) sugary drinks (carbonated soft drinks, fruit juices, sodas, and energy drinks), (3) snacks (packaged fatty or salty snacks), (4) ready-to-eat dishes (instant soups, ready-to-eat meals, and canned foods), (5) processed meats (sausage, meat derivatives, and cold cuts), (6) packaged baked goods (packaged bread products, pizza, and sandwiches), (7) delivery foods (fast foods and potato chips), and (8) cereals and energy bars (sugary breakfast cereals).

## 3. Results

The preliminary literature search, updated to December, 2021, yielded 1205 records. After excluding duplicates, 823 were considered potentially relevant, and were retained for the analysis of titles and abstracts. Then, 546 were excluded because they did not fit the attributes of the review approach or objective. After reviewing the full text of the remaining 277 records, only 28 met the inclusion criteria and were included in the final quantitative analysis [18,19,20,21,22,23,24,25,26,27,28,29,30,31,32,33,34,35,36,37,38,39,40,41,42,43,44,45]. Figure 1 shows a flowchart of the literature screening process.

Details of the design (cohort or cross-sectional), sample size (N) and sex ratio (%), minimum age or age range, setting (community or hospital), and country of the individual studies are shown in Table 2. The cross-sectional outnumbered the longitudinal design, whereas the recruitment settings were primarily community-based (100%, N = 28). The geographic distribution of studies spanned, in descending order of study prevalence, Europe (N = 19/28), America (N = 5), Asia (N = 3/28), and Africa (N = 1). Subjects were adults (18+) in most of the selected studies (23 of 28 studies, or 82%). Among all 123,608 subjects in the studies, gender was poorly balanced (approximately 33% males), and the geographic distribution of subjects ran through the following ascending order: 2970 African; 7132 Asian; 51,509 American; and 61,997 European. Concerning the representativeness of the sampling, the majority of studies used non-probability online surveys, i.e., river sampling, only two studies used telephone questionnaires [33,35], and just one survey enrolled participants within a school-based setting [44]. In total, 103 food and beverage entries belonging to the UPFs category, which were investigated in studies as single or food group consumption questions by the Food Frequency Questionnaire (FFQ), were counted. The most frequently examined UPF items were sweets (26 out of 103), sugary drinks (17 out of 103), and snacks (19 out of 103).

To optimize the summarizing of findings, conceptually similar UPFs entries were grouped into 8 categories. Thus, all data were reported as mean increased or decreased consumption for each UPFs category (Figure 2). Sweets (31.75% increase vs. 21.06% decrease), packaged baked goods (28.03% increase vs. 13.5% decrease), snacks (23.71% increase vs. 20.73% decrease), and breakfast cereals (13.45% increase vs. 3.5% decrease) showed a clear upward trend. However, only two studies focused on analyzing the latter food item [19,24]. By contrast, sugary drinks (14.55% increase vs. 22.65% decrease), ready-to-eat dishes (16.2% increase vs. 22.62% decrease), and delivery foods (10.6% increase vs. 31.43% decrease) showed a clear downward trend. No tendency was noticed for processed meat consumption (14.27% increase vs. 14.47% decrease). Figure 2 shows a graphic overview of the findings.

## 4. Discussion

We reviewed the existing literature on the consumption of UPFs in the context of the SARS-CoV-2 pandemic, focusing on studies that drew comparisons of consumption against pre-pandemic lockdown dietary habits. Our findings showed a clear upward trend for the consumption of sweets (chocolate, candies, cookies, pastries, cakes, desserts, and confectionery), snacks (packaged fatty or salty snacks), and bakery products (packaged bread products, pizza, and sandwiches), versus a decrease in ready-to-eat or delivery foods (instant soups, ready-to-eat meals, canned foods, fast foods, and potato chips), and sugary drinks (carbonated soft drinks, fruit juices, sodas, and energy drinks). No trend was observed for processed meat consumption. On this latter food group, a likely explanation for the lack of increase in consumption may lie in the fact that the COVID-19 pandemic affected meat production and the supply chain, causing a change in meat prices. Initially, prices of meat and meat products increased due to lower production and increased demand as a result of the panic. Subsequently, both production and demand for meat decreased significantly due to blockade restrictions and lower consumer purchasing power, which likely translates into lower meat consumption (and prices).

As regards the UPFs reported to have shown an increased consumption, the data require proper interpretation within a pandemic framework from a social and economic perspective. On one hand, affordability, much sought after during the SARS-CoV-2 health crash that placed many workers at risk of financial restrictions, was easily translated into choosing low-quality foods with a low nutritional value and typically high-calorie levels, saturated fat, trans fat, simple sugars, and sodium. Chocolate, sweets, candy, and snacks, also called “comfort foods”, fall more within this category of food items than any other. Furthermore, chocolate and candies are especially considered as comfort foods, and consumed most frequently in cases of depressive symptoms, since they can improve mood [46]. Precisely due to this positive effect on improving mood, it is easy to understand why their consumption intensified during the SARS-CoV-2 lockdown when social isolation and estrangement from relatives were already causing psychological distress, stress, and bad mood. Along this same line of thinking, nerve drinks, such as coffee and tea, were found to have an increased consumption during lockdowns [10]. However, the exceptional circumstances of the shutdown provided a positive opportunity for some people to spend more time cooking at home; though that attitude improved culinary approaches, it most often led to increased cooking of carbohydrate-dense foods, such as pizza, bread, and baked goods [9]. The desire to cook, especially foods that involve kneading, results in increased consumption of baked goods, which is generally not helpful in managing weight control during a pandemic and subsequent sedentary lifestyle. Furthermore, carbohydrate-dense foods are often tapped for their ability to compensate for physiological drops in blood sugar and serotonin. The increased availability of simple sugars, by raising insulin levels, induces greater storage of tryptophan within cells to synthesize serotonin.

As for UPFs showing a reduced consumption, we noticed a drop in ready-to-eat or delivery foods (instant soups, ready-to-eat meals, canned foods, fast foods, and potato chips), and sugary drinks (carbonated soft drinks, fruit juices, sodas, and energy drinks). To explain these findings against a pandemic background, consumers who spent more time in their homes opted to cook, and so, relied less on ready-to-eat dishes. Indeed, even though ready-to-eat food is popular overall, during the peak of the pandemic, there was a real shift toward cooking and baking at home, corroborating our findings on a reduced consumption of ready-to-eat meals and delivery foods. In reality, consumers of ready-to-eat meals usually tend to be financially wealthy and time-poor, but during lockdown, everyone had more time and, likely, less money.

Finally, the consumption of sugary drinks also decreased during lockdowns, reflecting the fact that most of them (alcoholic or not, carbonated or not) are commonly enjoyed in specific situations, such as parties, at movies, meals out, and so on [47]. However, due to the orders to stay home, social situations typically associated with sugary drink consumption were almost non-existent during periods of SARS-CoV-2 lockdowns [10,48]. As to energy drinks, in particular, which also fall in this category, the decline in consumption is likely attributable to the closure of gyms throughout the lockdown period, and hence, to reduced physical activity [49]. Moreover, it is not surprising that consumption was down, as these drinks carry insomnia and restlessness as side effects, which were already prevalent during the lockdown [50].

During the lockdown, an increase in the frequency of food intake has been reported, especially in the consumption of snacks and unhealthy foods. In addition, preliminary studies highlight increases in body weight and BMI since pandemic onset as linked to less healthy food choices [51]. Among these, increased consumption of UPFs may impact the overall health in terms of excess weight, visceral fat, and metabolic syndrome [52]. Furthermore, higher UPF consumption is linked to higher risks of cardiovascular, coronary, and cerebrovascular diseases [53]; a higher overall risk of cancer, including breast cancer [54]; and physical frailty in older adults [55]. The effects of a diet high in UPFs, and nutritionally unbalanced, can also impact the severity and outcomes of SARS-CoV-2 infection. Hospitalization, severe illness, and mortality from SARS-CoV-2 infection are all more likely in people with comorbidities such as obesity, diabetes, and hypertension [56].

Some limitations of the study should be considered. The small amount of data and heterogeneity of variables related to the different survey questionnaires used to investigate diet during the lockdown period lower the reliability of this summary in terms of qualitative consistency. Also, the European predominance of the selected studies left little room for a global view of consumption trends. Regarding representativeness of sampling, the majority of studies used online non-probability surveys, i.e., river sampling; as known, this sampling leads to coverage bias since not all subpopulations are proportionally represented, or even not at all in digital media. Lastly, although the cumulative examined population size was satisfactory, it should be noted that the sample size was extremely low, at fewer than 100 sample units for a minority group of three studies.

## 5. Conclusions

In this study, we gave an overview of existing data on changes in UPF consumption driven by SARS-CoV-2 lockdown to derive a synthesis hitherto lacking in the literature. The main findings were a clear upward trend in the consumption of sweets (chocolate, candy, cookies, pastries, cakes, desserts, and confectionery), snacks (packaged fatty or salty snacks), and baked goods (packaged bread, pizza, and sandwiches), versus a decrease in ready-to-eat and delivery foods (instant soups, canned foods, fast foods, and chips) and sugary drinks (carbonated beverages, juices, sodas, and energy drinks). No tendency was inferred for processed meat consumption.

In light of the well-known human health effect of UPF consumption in all age groups, and the continuing health emergency, knowledge on this topic needs to be further expanded and managed from a preventive perspective. These preliminary data may offer guidance in formulating specific dietary recommendations to better address this public health challenge.

## Figures and Tables

**Figure 1 foods-11-00950-f001:**
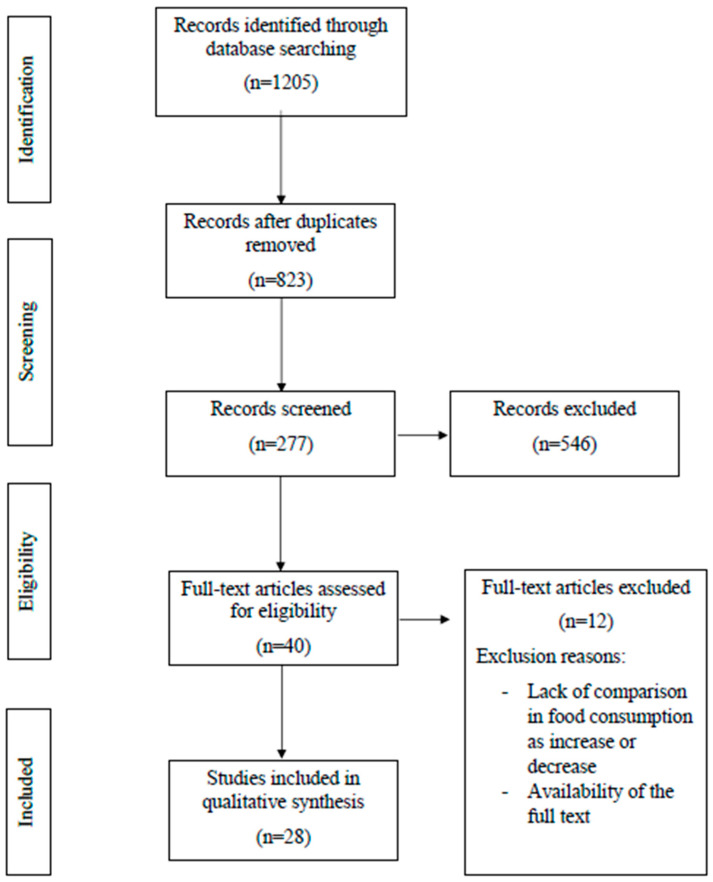
Flowchart of the literature screening process.

**Figure 2 foods-11-00950-f002:**
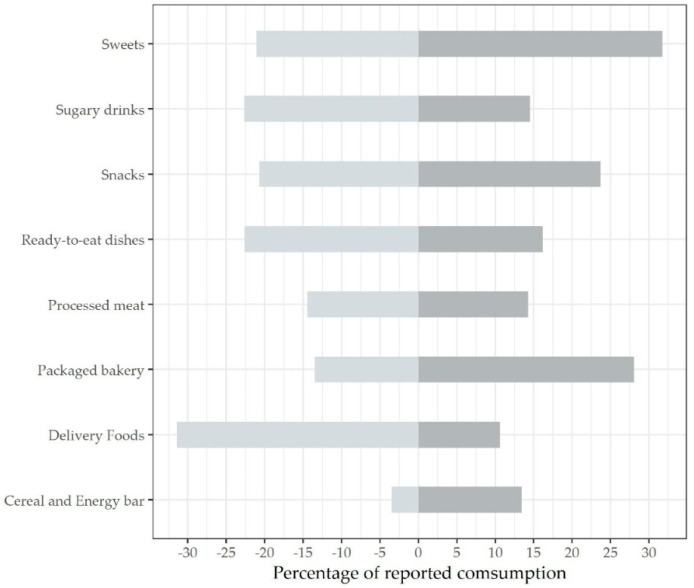
Percentage of reported consumption.

**Table 1 foods-11-00950-t001:** Search strategy used in the US National Library of Medicine (PubMed) and Medical Literature Analysis and Retrieval System Online (MEDLINE), and adapted to the other sources, according to selected descriptors.

Strategy	Descriptor Used
#1	(Diet*) OR (Dietary Lifestyle*) OR (Eating habit*) OR (Food intake*) OR (Dietary habit*) OR (Eating habit*) OR (Dietary behavior*) OR (Dietary pattern*) OR (Habit*) OR (Food*) OR (Beverage*) OR (Snack*) OR (Processed food*) OR (Sweet*) OR (Soft drink*) OR (soda*) OR (canned food*) OR (processed meat*) OR (fast food*)
#2	(Change*) OR (Difference*) OR (Different*) OR (Modification*)
#3	(COVID 19) OR (SARS-CoV-2) OR (Coronavirus)
#4	(Review) OR (Systematic review) OR (Narrative review) OR (Meta-analysis)
#5	#1 AND #2 AND #3 NOT #4

**Table 2 foods-11-00950-t002:** Selected studies investigating changes in ultra-processed food consumption during the period of SARS-CoV-2 lockdown compared with previous habits (N = 28).

Author, Year (Ref.)	UPF Diet Exposure	Diet Assessment Tool	Design	N	Sex (%)	Age (Years)	Setting	Country	Results	Summary of Findings
Luo Y et al., 2020 [18]	Snacks Sugary drinks	Questionnaire	Cross-sectional	2272	18.3% (M) 81.7% (F)	18+	Community	Asia (China)	Reduction in snack consumption (−23.6%) and in sugary drinks (−26.6%)	↓ sugary drinks ↓ snacks
Deschasaux-Tanguy M et al., 2020 [19]	Processed meat Plant-based steaks and soja based-steaks Ready-made dishes canned Sandwiches/pizza/savory pies Sweets and chocolate Cookies and cakes Fruit juice Sugary drinks and soda	Questionnaire	Cross-sectional	37,252	47.7% (M) 52.3% (F)	18+	Community	Europe (France)	Increased consumption of breakfast cereals (+4.6%, −3.5%); sweets and chocolate (reported by +21.7%, −9%) of the participants); cookies and cakes (+20.4%, −10%); fruit juice (+6.2%, −5.5%); and decreased consumption of sandwiches, pizzas, or savory pies (+5.9%, −17.4%); processed meat (+8.4%, −14.7%); plant−based steaks and soja−based steaks (+2.8%, −3.7%); ready−made dishes (+2.7%, −4.8%); sugary drinks and sodas (+3.7%, −4.8%)	↑ breakfast cereals ↑ sweets and chocolate ↑ cookies and cakes ↑ fruit juice ↓ sandwiches ↓ pizzas and savory pies ↓ processed meat ↓ plant-based steaks and soja-based steaks ↓ ready-made dishes ↓ sugary drinks and sodas
Di Renzo L et al., 2020 [20]	Salted snacks Sweet beverages Processed meat Delivery foods Packaging sweets	Questionnaire	Cross-sectional	977	23.9% (M) 76.1% (F)	12+	Community	Europe (Italy)	Decreased consumption of salty snacks (+9%, −12%), sweet beverages(+5%, −8%), processed meats (+3%, −6%), delivery foods (+2%, −20%), and confectionery packaging (+11%, −15%)	↓ salted snacks ↓ sweet beverages ↓ processed meat ↓ delivery foods ↓ packaging sweets
Malta DC et al., 2020 [21]	Savory snacks Chocolate/sweet Biscuits/pieces of tart	Questionnaire	Cross-sectional	45,161	46.4% (M) 53.6% (F)	18+	Community	America (Brazil)	Increase in the number of people consuming chocolate/sweet cookies/tart pieces (41.3% to 47.1%, i.e., +5.8%), savory snacks (9.5% to 13.2%, i.e., +3.7%) more than two days per week.	↑ Savory snacks ↑ Chocolate/sweet ↑ Biscuits/pieces of tart
Skotnicka M et al., 2021 [22]	Canned food Sweets and snacks Juices and sweet drinks	Questionnaire	Cross-sectional	1071	43.6% (M) 56.4% (F)	18+	Community	Europe (Poland, Austria, UK)	Increase in consumption of canned food (6.54% before, 10.08% after, i.e., +3.54%), sweets and snacks (16.06% before, 21.67% during, i.e., +5.61%). Decrease in consumption of juices and sweet drinks (16.90% before, 16.25% after, i.e., −0.65%)	↑canned food ↑sweets and snacks ↓juices and sweet drinks
Celorio-Sardà R et al., 2021 [23]	Processed meat Industrial pastries Chocolate Salty snacks Soft drinks	Questionnaire	Cross-sectional	321	20.2% (M) 79.8% (F)	18+	Community	Europe (Spain)	There was an increase in consumption of processed meat (+20%, −16%), chocolate (+28%, −14%), salty snacks (+28%, −10%), and industrial confectionery (+20%, −18%). Soft drink consumption generally decreased (−13%, +11%)	↑Processed meat ↑Chocolate ↑Salty snacks ↑Industrial confectionery ↓Soft drink
Bin Zarah A et al., 2020 [24]	Sweets Potato chips and salty snacks Cold breakfast cereals Processed meat Fruit juice Sweet beverages	Questionnaire	Cross-sectional	3101	19.8% (M) 79.4% (F)	18+	Community	America (USA)	Increased consumption of sweets (+43.8%), potato chips and salty snacks (+37.4%), cold breakfast cereals (+22.3%), processed meat (+20%), fruit juice (+11.7%), sweet beverages (+10.6%)	↑Sweets ↑Potato chips and salty snacks ↑Cold breakfast cereals ↑Processed meat ↑Fruit juice ↑Sweet beverages
Fanelli RM, 2021 [25]	Canned products Sweet snacks	Questionnaire	Cross-sectional	50	30% (M) 70% (F)	18+	Community	Europe (Italy)	Increased consumption of canned products (+31%, −26%) and sweet snacks (+41%, −25%)	↑Canned products ↑Snacks
Dobrowolski H et al., 2021 [26]	Sugary products and sweets Fast food, salty snacks, and sweet drinks	Questionnaire	Cross-sectional	183	21.9% (M) 78.1% (F)	17-71	Community	Europe (Poland)	Increased consumption of sugary products and sweets (+36.2%, −18.6%), and fast food, salty snacks and sweet drinks (+32%, −28%)	↑Sugary products and sweets ↑Fast food, salty snacks, and sweet drinks
Sulejmani E et al., 2021 [27]	Non-homemade sweets Sweet drinks	Questionnaire	Cross-sectional	689	29.2% (M) 70.8% (F)	18+	Community	Europe (Kosovo)	Consumption of non−homemade sweets increased (35% reported an increase and 22% a decrease, i.e., +13%) whereas consumption of sweet drinks decreased (23% reported an increase and 37% a decrease, i.e., −14%)	↑ Non-homemade sweets ↓Sweet drinks
Yang G-Y et al., 2021 [28]	Snacks	Questionnaire	Cross-sectional	2723	29.3% (M) 70.7% (F)	18+	Community	Asia (China)	Snack consumption had increased in 38.2% of the participants, and decreased in 13.6%, i.e., +24.6%	↑Snacks
Grant F et al., 2021 [29]	Sugary drinks Sweets and pastries	Questionnaire	Cross-sectional	2768	48.2% (M) 51.8% (F)	18+	Community	Europe (Italy)	Reduced consumption of sugary drinks (16.3% vs. 5.3%, i.e., −11%). Increased consumption of sweets and pastries (36.9% vs. 12.3%, i.e., +24.6%)	↓Sugary drinks ↑Sweets and pastries
Pujia R et al., 2021 [30]	Chocolate Sweet packaged snacks Ice cream and desserts Pizza and bakery products Sweetened beverages Candies	Questionnaire	Cross-sectional	439	56% (M) 44% (F)	5 to 14	Community	Europe (Italy)	Increase in the consumption of chocolate (32%), sweet packaged snacks (34%), ice cream and dessert (32%), pizza and bakery products (47%) was found. Decrease in the consumption of sweetened beverages (23%) and candies (29%)	↑Chocolate ↑Sweet packaged snacks ↑Ice cream and desserts ↑Pizza and bakery products ↓Sweetened beverages ↓Candies
Cheikh Ismail L et al., 2020 [31]	Ready-to-eat meals Fast food	Questionnaire	Cross-sectional	2970	28.4% (M) 71.6% (F)	18+	Community	Africa (Greater Middle East regions)	Ready−to−eat meals decreased by 0.9% and fast food by 23.5%	↓Ready-to-eat meals ↓Fast food
Sinisterra Loaiza LI et al., 2020 [32]	Cold meat and sausage Sugary drinks Salty snacks Pizza and hamburgers Sweets Fruit juice	Questionnaire	Cross-sectional	1127	30% (M) 70% (F)	18+	Community	Europe (Spain, Galizia)	Decreased consumption of cold meat and sausage (−29.6%), sugary drinks (−21.2%), salty snacks (−26.8%), and pizza and hamburgers (−48.7%). Increased consumption of sweets (+36.4%) and fruit juice (+15.2%)	↓Cold meat and sausage ↓Sugary drinks ↓Salty snacks ↓Pizza and hamburgers ↑Sweets ↑Fruit juice
Ruiz-Roso MB et al., 2020 [33]	Sugary foods Snacks	Questionnaire	Cross-sectional	72	48.6% (M) 51.4% (F)	45–77	Community	Europe (Spain)	Consumption of sugary foods increased from 2.9% to 5.7%, (+2.8%), as did consumption of snacks from 5.7% to 12.9%, (+7.2%)	↑Sugary foods ↑Snacks
Grabia M et al., 2020 [34]	fast food, convenience food, salty snacks, sweet snacks	Questionnaire	Cross-sectional	124	17% (M) 83% (F)	17–45 years	Community	Europe (Poland)	Decreased consumption of fast food (−32%, +14%), convenience foods (−29%, +16%), salty snacks (−29%, +19%), sweet snacks (−22%, +21%), energy drinks (−15%, +13%), and increased consumption of sweet drinks (−11%, +19%)	↓Fast food ↓Convenience foods ↓Salty snacks ↓Sweet snacks ↓Energy drinks ↑Sweet drinks
Bonaccio M et al., 2021 [35]	Pizza Cookies Chocolate Fruit drinks Savory snacks Soft drinks Vegetable meat substitutes Vegetable cheese substitutes	Questionnaire	Cross-sectional	2992	40.4% (M) 59.6% (F)	18+	Community	Europe (Italy)	Increased consumption of pizza (+31.2, −9.6%), cookies (+18%, −6.2%), chocolate (+18.6%, −7.5%). Decrease in consumption of fruit drinks (+4.2% vs. −7.9%), savory snacks (+7.5%, −12.5%), soft drinks (+4.7%, −12.3%), vegetable meat substitutes (+1.2%, −10.2%), vegetable cheese substitutes (+0.8%, −10.4%)	↑Pizza ↑Cookies ↑Chocolate ↓Fruit drinks ↓Savory snacks ↓Soft drinks ↓Vegetable meat substitutes ↓Vegetable cheese substitutes
Davila-Torres DM et al., 2021 [36]	Chocolate and candies Cookies Meat derivates Preserves Packaged juice	Questionnaire	Cross-sectional	74	56.84% (M) 43.24% (F)	19–32	Community	America (Perù)	Increased consumption of chocolate and candy (+5.7%), cookies (+7.7%), meat derivatives (+14.1%), canned goods (+4.4%), packaged juices (+13.8%), sugar−sweetened beverages (+13.2%), and energy drinks (+ 4.4%)	↑Chocolate and candy ↑Cookies ↑Meat derivatives ↑Canned goods ↑Packaged juices ↑Sugar-sweetened beverages ↑Energy drinks
Silva MN et al., 2021 [37]	Fruit juice Soft drinks Savory snack Sweet snacks Delivery meals Ready meals Canned foods	Questionnaire	Cross-sectional	5856	NA	16+	Community	Europe (Portugal)	Decrease in consumption of fruit juice (+12.3%, −15.6%), soft drinks (+3.7%, −32.8%), savory snacks (+8.9%, −29.5%), delivery meals (+7.5%, −43.8%), ready meals (+4.9%, −40.7%), and canned foods (+9.7%, −15.2%). Increased consumption of sweet snacks (+30.9%, −20%)	↓Fruit juice ↓Soft drinks ↓Savory snack ↑Sweet snacks ↓Delivery meals ↓Ready meals ↓Canned foods
Sánchez-Sánchez E et al., 2021 [38]	Snacks and jellybeans Soft drinks	Questionnaire	Cross-sectional	637	25.14% (M) 74.94% (F)	18+	Community	Europe (Spain)	Increased consumption of snacks and jellybeans (+14%), and soft drinks (+9.3%)	↑Snacks and jellybeans ↑Soft drinks
Giacalone D et al., 2020 [39]	Carbohydrate drinks Pastries Fast food	Questionnaire	Cross-sectional	2462	28.9% (M) 71.7% (F)	18+	Community	Europe (Denmark)	Increased consumption of carbohydrate drinks (−5.6%, +21.4%), pastries (−18.4%, +21.1%), and decreased consumption of fast food (−25.4%, +15.1%)	↑Carbohydrate drinks ↑Pastries ↓Fast food
Pertuz-Cruz SL et al., 2021 [40]	Sugary beverages Pastries	Questionnaire	Cross-sectional	2745	26.8% (M) 73.1% (F)	18+	Community	America (Colombia)	Increased consumption of sugary beverages (+18%, −7%) and pastries (+27%, −25%)	↑Sugary beverages ↑Pastries
Scarmozzino F et al., 2020 [41]	Sweet food and chocolate Salty snacks	Questionnaire	Cross-sectional	1929	33% (M) 67% (F)	18+	Community	Europe (Italy)	Increased consumption of sweet food and chocolate (+42.5%, −13.5%), and salty snacks (+23.5%, −18.9%)	↑Sweet food and chocolate ↑Salty snacks
Galali Y, 2021 [42]	Snacks Energy drinks Sweets Juice Canned fish Processed meat Delivery food	Questionnaire	Cross-sectional	2137	43.4% (M) 56.6% (F)	0+	Community	Asia (Iraq)	Increase consumption of snacks (+20.4%, −11%). Decreased consumption of energy drinks (+11.7%, −23.7%), sweets (+4.3%, −40.6%), juice (+4.3%, −40.6%), canned fish (+3.6%, −22.3%), processed meat (+2.4%, −31.6%), and delivery food (+2.1%, −44.6%)	↑Snacks ↓Energy drinks ↓Sweets ↓Juice ↓Canned fish ↓Processed meat ↓Delivery food
Romeo-Arroyo E et al., 2020 [43]	Sausages and cold cuts Sweets	Questionnaire	Cross-sectional	600	49.9% (M) 50.1% (F)	18+	Community	Europe (Spain)	Increased consumption of sausages and cold cuts (32%, −16%), and sweets (50%, −15%)	↑Sausages and cold cuts ↑Sweets
Skolmowska D et al., 2021 [44]	Pastries and cakes Delivery meals Sausages	Questionnaire	Cross-sectional	2448	NA	15–20	Community	Europe (Poland)	Decreased consumption of pastries and cakes (−11.1%), delivery meals (−5.9%), and sausages (−2.5%)	↓Pastries and cakes ↓Delivery meals ↓Sausages
Sadler JR et al., 2021 [45]	Sweets Savory snacks Fast foods	Questionnaire	Cross-sectional	428	36.9% (M) 63.1% (F)	18+	Community	America (USA)	Increased consumption of sweets (+40.9%, −34.8%) and savory snacks (+33.6%, −22.9%), and decreased consumption of fast foods (+22.9%, −39%)	↑Sweets ↑Savory snacks ↓Fast foods

## Data Availability

The original contributions presented in the study are included in the article; further inquiries can be directed to the corresponding author.

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
