# Peer review of "Public Health Response to the SARS-CoV-2 Pandemic: Concern about Ultra-Processed Food Consumption"

_foods, 2022, doi:10.3390/foods11070950_

Round 1

Reviewer 1 Report

Reviewer(s)' Comments to Author:

Recommendation: Minor Revision

Manuscript Foods- 1628207 with title “Public Health Response to the Sars-Cov-2 Pandemic: Concern about Ultra-Processed Food Consumption”

Recent literature points to troubling evidence that policies adopted to address the SARS-CoV-2 pandemic may have contributed to divert eating habits toward a poorer diet like ultra-processed foods (UPFs).This study explored consumption of UPFs as a descriptor of an unhealthy diet in SARS-CoV-2 pandemic period. The term UPF derives from the classification, widely used in the literature, which divided foods into four categories according to the level of processing to which they are subjected. The results of this study show increased consumption of UPFs, especially sweets, snacks, and baked goods in the examined period.

Comments:

Abstract

  1. Please better clarify the purpose of the study.
  2. Please revise your findings and originality of the research.

Methodology:

Please better clarify data collection

Please provide the sampling inclusion criteria and exclusion criteria

Results:

There are too many tables. Perhaps consider linking them or showing them more clearly or showing them in Appendix, especially such as Table 2.

Conclusions

Please write a clearer conclusion. Revise your findings and originality of the research and should provide a clear scientific justification for the study.

Literature

The manuscript shows the relevant references related to this study.

Author Response

File attached.

Reviewer 2 Report

The study investigated how ultra-processed food consumption changed during the COVID pandemic. Overall, the paper is well-written and the objectives, methods and results are well explained. The authors may want to expand the discussion to explain some of the findings.

Comments:

Line 156- What is meant by “cross-checked data and fast food”?

Line 248- Why did processed meat consumption not increase? Do the authors have an explanation for this result?

Line 261- What are nerve drinks?

Line 264- Why do the authors think consumption of carbohydrate dense foods increased?

Line 280- Was there an increased consumption of alcoholic sugary drink? If they were separated from the sugary drink category.

Author Response

File attached.
